# LEARNABILITY AND EXPRESSIVENESS IN SELF-SUPERVISED LEARNING

## ABSTRACT

We argue that representations induced by self-supervised learning (SSL) methods should both be *expressive* and *learnable*. To measure expressiveness, we propose to use the Intrinsic Dimension (ID) of the dataset in representation space. Inspired by the human study of Laina et al. (2020), we introduce Cluster Learnability (CL), defined in terms of the learning speed of a KNN classifier trained to predict $K$-means cluster labels for held-out representations. By collecting 30 state-of-art checkpoints, both supervised and self-supervised, using different architectures, we show that ID and CL can be combined to predict downstream classification performance better than the existing techniques based on contrastive losses or pretext tasks, while having no requirements on data augmentation, model architecture or human labels. To further demonstrate the utility of our framework, we propose modifying DeepCluster (Caron et al., 2018) to improve the learnability of the representations. Using our modification, we are able to outperform DeepCluster on both STL10 and ImageNet benchmarks. The performance of intermediate checkpoints can also be well predicted under our framework, suggesting the possibility of developing new SSL algorithms without labels.

## 1 INTRODUCTION

We consider the unsupervised learning scenario, which aims at learning distributed representations for complex natural data without human annotations. Despite great recent progress in self-supervised learning (Chen et al., 2020a; Caron et al., 2021; Grill et al., 2020; Caron et al., 2020; 2018; He et al., 2020; Chen and He, 2021), the problem of properly evaluating the quality of the learned representations has not been fully explored. We approach this question by drawing an analogy with the evolution of human language. Recent literature (Swarup and Gasser, 2008; Smith et al., 2013) suggests that linguistic structure arises as a trade-off between *expressiveness* and *learnability*. On one hand, expressiveness is the ability to discriminate objects and concepts of the world. On the other hand, Kirby et al. (2014) states that language evolution undergoes an iterated learning process, where each individual acquires language by observing a limited amount of language outputs of others (Chomsky, 1980). This process induces a form of compression, which makes the language simpler and more learnable for the next generation. Linguistic structure, e.g., compositionality, emerges between these opposing forces.

A similar guiding principle can be applied when learning representations of natural data. Many existing methods in self-supervised learning can be considered as attempts to increase expressiveness of the representations, i.e., by maximizing or controlling the mutual information between inputs and representations, either through reconstruction (Vincent et al., 2008; Higgins et al., 2017) or contrastive learning (Oord et al., 2018; Bachman et al., 2019b). We argue that in addition to maximizing expressiveness, a good representation should also be learnable, e.g. support generalization from few examples (Laina et al., 2020).

In this paper, we propose an evaluation framework for self-supervised learning relying on expressiveness and learnability. Compared with previous work (e.g., Wang and Isola, 2020), our framework is independent of (i) the availability of downstream task labels or predefined data augmentation techniques, and (ii) architecture choices and dimensionality of the latent representations, which makes it applicable to all previous SSL methods. We propose to measure expressiveness in terms estimators of the intrinsic dimension (ID) of the data representations (Ansuini et al., 2019). Taking inspiration

from Laina et al. (2020), we quantify learnability in terms of the learning speed (in terms of number of examples) of a KNN learner trained to predict the $K$-Means cluster labels for held out representations. We dub this method Cluster Learnability (CL). We show that high ID and CL correlate with high downstream performance for 30 different pretrained checkpoints across different architectures.

To further demonstrate the usefulness of our framework, we present a case-study in which we improve upon DeepCluster (Caron et al., 2018), which is a self-supervised learning method that iterates between clustering the representations and fitting the cluster labels. Inspired by our CL measure, we refine the cluster labels by executing KNN at the end of each $K$-means clustering round. The resulting method, KNN-DeepCluster, is able to outperform DeepCluster in both ImageNet and STL10. When visualizing the training trajectories on the CL-ID plane, we find that our method indeed improves learnability at convergence. The correlation between performance and learnability holds also for checkpoints obtained during training, which suggests the possibility of using our framework to guide the development of new SSL algorithm, without labels at hand.

Our contributions are summarized below:

- We analyze self-supervised learning through the lens of expressiveness and learnability, and we propose tractable way to estimate them with Intrinsic Dimension (ID) and Cluster Learnability (CL).

- We carry out a large-scale evaluation on ImageNet and STL10 with 30 state-of-the-art self-supervised and supervised checkpoints. We show that CL-ID is more correlated with the Top1 accuracy than previous approaches, with a Pearson's $r$ of 0.93.

- We propose a modification to DeepCluster to increase its cluster learnability. The resulting algorithm, KNN-DeepCluster is able to outperform DeepCluster.

## 2    RELATED WORK

**Representation Evaluation**    Several recent works address the question of representation evaluation in self-supervised learning. Whitney et al. (2021) propose to use the learning dynamic of the downstream classifiers to measure the representation complexity, however the method depends on extra human labels. Wang and Isola (2020) decompose the contrastive loss into an alignment loss and a uniformity loss, which can be used to analyze the representation. However, their method requires access to the projection head output, limiting their applicability to general architecture. Pretext tasks like jigsaw, rotation prediction are shown to be well correlated with the supervised evaluation (Reed et al., 2021; Deng and Zheng, 2021) or architecture search (Liu et al., 2020). However, as is shown in our experiments, they only provide limited predictive power when applying to a large pretrained family of checkpoints including both supervised and self-supervised models. In addition, these methods depends on predefined data augmentation, which might be task specific.

**Manifold Intrinsic Dimension**    Intrinsic dimension can be thought of as the smallest number of variable needed to approximate the representation manifold. Note that it is different from the intrinsic dimension of loss landscape (Li et al., 2018), which measures minimum dimension of parameter space needed to optimize. We build upon extensive work on estimating intrinsic dimension. Applying local neighborhood information to estimate the intrinsic dimension is not a new idea, and it is shown to be more superior to the global eigenvalue approach (Pettis et al., 1979b). Ansuini et al. (2019) apply the TwoNN estimator (Facco et al., 2017) to the non-linear representation manifold of modern deep neural nets. They find that the intrinsic dimension is inversely correlated to the classification accuracy, and that the deep layers have smaller intrinsic dimensions. Their work is further extended to confirm that natural images lies in a low-dimension manifold (Pope et al., 2021), and lower ID datasets leads to better generalization. Our work presents a more nuanced view about ID. It suggests that in order to learning representation from scratch, a least amount of ID is needed to convey the information from the dataset, e.g., a constant-value representation has an ID of 0, but no capacity to generalize at all. More discussion can be found in Section 5.

**Learnability, Ease-of-Transmission and Compression**    Learnability has been argued to be a hallmark of the human language in order to be effortlessly transmitted through generations (Kirby et al., 2014; Rafferty et al., 2011; Beckner et al., 2017; Zhou and Yurovsky, 2021; Kampen, 2004), and

it is also true for visual concepts like color (Xu et al., 2010), categories (Griffiths et al., 2006), shapes (Portelance et al., 2021) etc. In deep learning, it has been explored in the context of emergent communication (Ren et al., 2020; Guo et al., 2019; Li and Bowling, 2019), language drift (Lu et al., 2020), and neural module networks (Vani et al., 2021), but it is less explored for vision representation learning, except for a human study on just two SSL methods Laina et al. (2020). Learnability has a tight connection to compression (Chaitin, 2007) and prequential codelength (Dawid, 1984), which quantifies the compression levels with the online learning error. Existing works (Blier and Ollivier, 2018) has use it to support the generalization ability of the learner (e.g., deep neural nets) on the dataset (e.g., labeled images). However, we use it to quantify the learnability of the representation, in the sense that if the emerged Kmeans clustering is more learnable, then the same KNN learner could achieve a lower compression bound via prequential coding.

## 3 PROPOSED FRAMEWORK

In this section, we introduce our setup and define our measures of expressiveness and learnability.

### 3.1 NOTATION

We consider the following setting: we assume we have a dataset of images $\{x_i\}_{i=1}^{N}$ represented in a pixel space $\mathcal{X} \subset \mathbb{R}^d$, where $x_i \sim \mathcal{P}$ are sampled i.i.d from some natural image distribution $\mathcal{P}$ over $\mathcal{X}$. We also consider *representation maps* $\mathcal{F} : \mathcal{X} \to \mathbb{R}^m$, typically pretrained neural networks, which represent any image as an $m$-dimensional vector. Any such map forms a representation dataset $Z = \{z_i\}_{i=1}^{N}$ where $z_i = \mathcal{F}(x_i)$. Given $z_i \in Z$ and an integer $k \geq 1$, we denote by $NN(z_i, k)$ the $k$-th nearest neighbor of $z_i$. While we generally mean 'nearest' with respect to Euclidean distance, in practice we will also use the cosine distance function,[1]

$$D(z_1, z_2) = 2 - 2\cos(z_1, z_2) \tag{1}$$

Let $r_{ik} = D(z_i, NN(z_i, k))$ be the distance of the data point $z_i$ to its $k$-th nearest neighbor.

### 3.2 INTRINSIC DIMENSION (ID)

To measure expressiveness, we propose to use the notion of intrinsic dimensionality (ID) of the data in the representation space (Pettis et al., 1979a). Intuitively, as the representation becomes more expressive, we expect it to be able to represent images coming from more fine-grained categories, which could have multiple disentangled features such as color, shape, positions in space, etc. If these disentangled features correspond to different directions in some non-linear manifold, ID should yield a good surrogate for the expressiveness of the representation.

Inferring the intrinsic dimension of a highly nonlinear manifold is a challenging problem (e.g., Levina and Bickel, 2005). In this work, we leverage the nearest neighbor-based method of Facco et al. (2017) to estimate ID, which uses only the first two nearest neighbors of each point. This estimator (TwoNN) is shown to be reliable with respect to representation dimensions and scalable to real-world datasets with deep neural networks (Ansuini et al., 2019). Assuming the data density is approximately constant around data points up to the distance of their neighbors, this method exploits the simple relationship between the distribution of the ratio of first and second nearest neighbor distances and the intrinsic dimension.

Formally, let $\mu_i = r_{i2}/r_{i1}$ be the ratio of distances for the data point $z_i$. We suppose the points are sampled on a manifold with intrinsic dimension $d$. It can be shown that, under the above local uniformity assumption, $\mu_i$ follows a Pareto distribution with parameter $d + 1$ on $[1, \infty)$. That is

$$P(\mu_1, \cdots, \mu_N | d) = d^N \prod_{i=1}^{N} \mu_i^{-(d+1)} \tag{2}$$

While $d$ can be computed by maximizing the likelihood, we follow a much simpler method proposed by Ansuini et al. (2019) and estimate $d$ with a linear regression on the empirical cumulate of the distribution of $\mu$, which is shown to have good practical performance (Facco et al., 2017). This estimator is exact for uniformly distributed data. It is also shown to remain asymptotically correct for non-uniform sampling of $z_i$ when the image dataset is large enough (Ansuini et al., 2019).

---

[1] Both distances produce similar results in our experiments.

### 3.3 Cluster Learnability (CL)

To measure the representation learnability, we take inspiration from Laina et al. (2020). In this work, it is shown that the classes obtained by clustering the representations of pretrained self-supervised models can be easily learned by humans. We propose a *cluster learnability* (CL) metric as a way to automate their procedure and measure the learnability of the representation.

Let $\{z_i, \tilde{y}_i\}_{i=1}^N$ be the labelled dataset obtained by clustering the representation (e.g., with $K$-means). We want to define the learnability of the representation as the learning speed of a KNN classifier with this labelled dataset. To formalize this, we use a prequential approach (Dawid, 1984). Let $\hat{y}_i = KNN(z_i|\{z_{k<i}, \tilde{y}_{k<i}\})$ denote the prediction of the $i$-th label after seeing the pairs $(z_k, \tilde{y}_k)$ for $k < i$. Learnability is defined as the online learning accuracy of our KNN classifier:

$$CL = \frac{1}{N} \sum_{i=1}^N [\hat{y}_i == \tilde{y}_i]$$

(3)

This metric can be connected to the prequential codelength of the cluster assignment dataset where the conditional model is a formed by a KNN classifier. We discusss this further in Appendix B.

## 4 Experiment: Empirical Evaluation of the CL-ID Framework

In this section, to assess the main hypothesis of the paper and the relevance of our metrics, we perform experiments designed to answer the following questions:

**Q1:** Are high-performing models consistently both expressive and learnable?

**Q2:** Can our CL-ID framework robustly predict the model performance better than the existing SSL evaluation frameworks?

### 4.1 Setup

We select 11 different self-supervised checkpoints and download the official ImageNet checkpoints. We include checkpoints with different training epochs and different architecture. Combining the supervised checkpoints results in 30 checkpoints in total. A complete list can be found in Table 3 in the appendix. We mainly use the KNN evaluation on the validation data using the ground-truth labels to measure the performance of the model, which has been shown to be well correlated with the linear evaluation but computationally less expensive (Caron et al., 2021). CL is computed on the train set to have a better estimate, while the ID is computed on the validation. For STL-10, we use the unlabeled split, which has more images.

In our experiments, we use 1000 clusters for ImageNet and 320 for STL-10, which are approximately the square root of the dataset size. We report results with 1 neighbor for our KNN learner. We normalize the features and use cosine distance for the K-means clustering and KNN learner. Other configurations of cluster numbers and neighbor numbers are also explored in section 4.4.

### 4.2 Qualitative Evaluation

We conduct a qualitative evaluation (Figure 1) to address **Q1**. We choose Alignment vs. Uniformity (Align-Unif) (Wang and Isola, 2020) as our baseline, since this work also does not depend on downstream labels. We compute both alignment loss and uniformity loss on the validation data.

On both dataset, the proposed CL-ID framework show that good representations tend have high cluster CL as well as high ID, forming clusters on the upper-right Pareto frontier. In contrast, the Align-Unif fails to show the results with a lower alignment and higher uniformity for good representations. We hypothesize that this is because Align-Unif framework is derived from the contrastive loss and therefore requires access to the post-projection output. However, not every checkpoint has the projection layer. Our results suggest that the Align-Unif framework is less flexible than ours and might not be applicable to the raw outputs coming from a pretrained backbone.

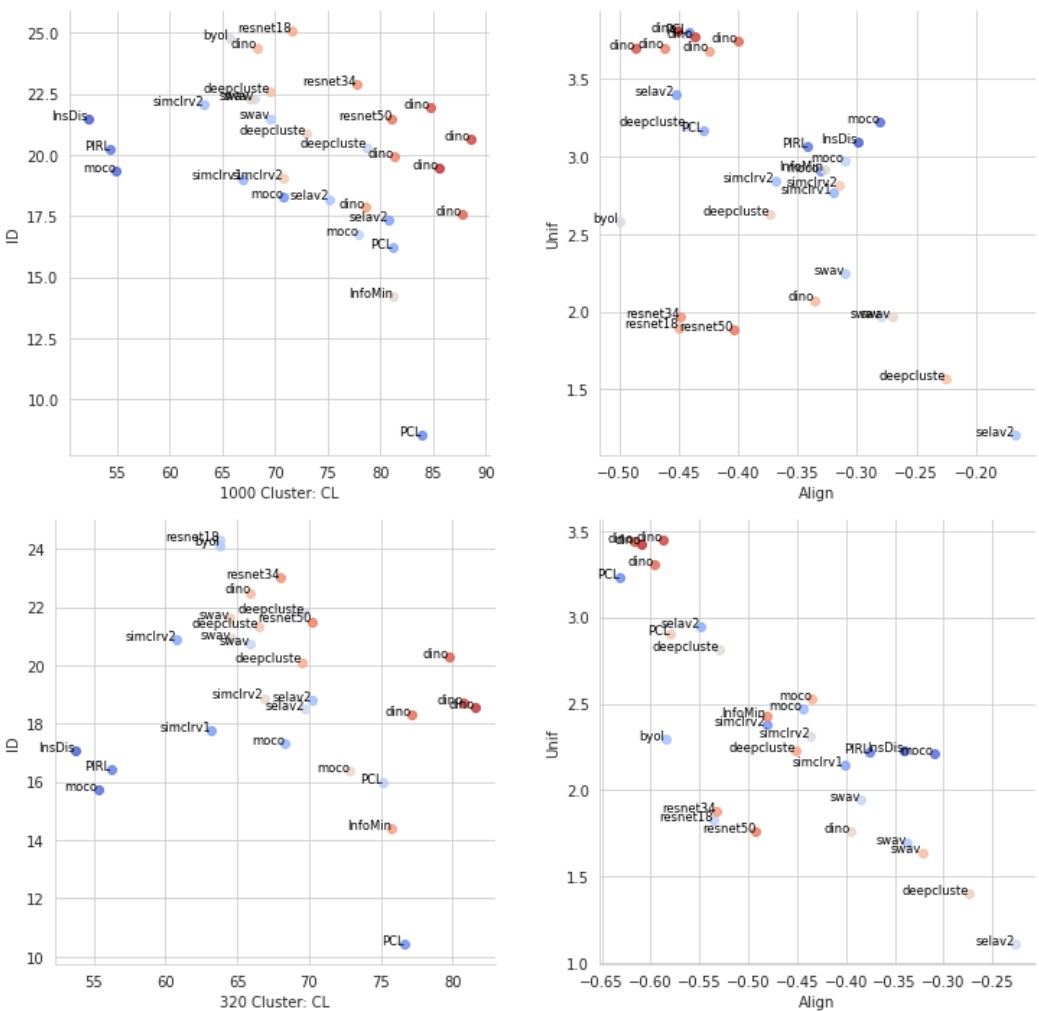

Figure 1: Visualization of SSL checkpoints for ImageNet (**first row**) and STL-10 (**second row**) on different frameworks. Color is used to show the KNN Top1 accuracies. Red indicates high accuracy, while blue indicates low accuracy. **Left Column** is our proposed CL-ID framework while **Right Column** is Align-Unif framework (Wang and Isola, 2020). Our framework shows qualitatively consistent results: Good representations tend have high CL as well as high ID (in the upper-right corner).

### 4.3 QUANTITATIVE EVALUATION

We also perform a quantitative evaluation (Figure 2) to address **Q1** and **Q2** by comparing the Top1 accuracy predictors from different framework. We come up with two Top1 accuracy predictors for our method:

$$\text{Vanilla CL-ID Predictor}: \quad CL + ID$$
$$\text{Linear CL-ID Predictor}: \quad K_1 \cdot CL + K_2 \cdot ID + K_3$$

where $K_1, K_2, K_3$ are obtained by linear regression with our collected 30 checkpoints.

The first baseline is linear Align-Unif Predictor, which is the linear regression of the alignment loss and uniformity loss. It is also learnt from the collected checkpoints.

$$\text{Linear Align-Unif Predictor}: \quad K_1 \cdot Align + K_2 \cdot Unif + K_3.$$

The second baseline is rotation prediction, which is one of the best pretext task proposed proposed by Reed et al. (2021). We randomly rotate the training images with $0, 90, 180, 270$ degrees, train

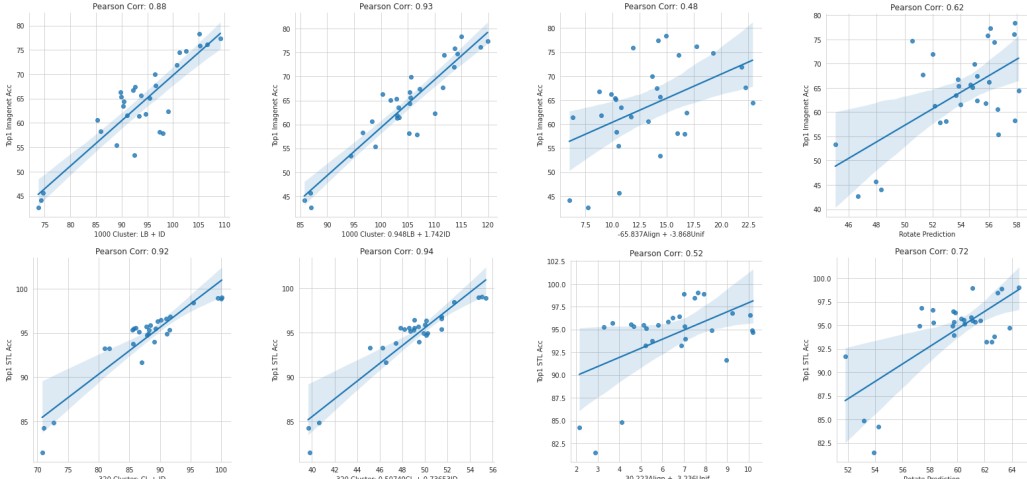

Figure 2: Correlation Plots between Top1 accuracy and its various predictors. The first row is ImageNet and the second row is STL-10. **First Column:** Vanilla CL-ID Predictor. **Second Column:** Linear CL-ID Predictor. **Third Column:** Linear Align-Unif Predictor (Wang and Isola, 2020). **Fourth Column:** Rotation Prediction (Reed et al., 2021). Our predictor has a higher correlation to the Top1 image classification accuracy than the baselines even with the vanilla version.

a KNN classifier on the train images to predict a 4-way classification, and then report the rotation prediction accuracy on the validation images.

Our results show that, for both datasets, our linear predictor beats our vanilla one, which already beats all the other methods in terms of the correlation scores. While the linear Align-Unif predictor achieves about 0.5 correlation, the learnt parameters $K_1, K_2$ are negative, which support our observation in Figure 1 that alignment-uniformity framework does not produce qualitatively correct results. The pretext tasks like rotation prediction have certain correlation, but it is not as good as our methods when applied to a wider range of SSL methods and it is data augmentation heavy.

We also tried combining alignment loss, uniformity loss and rotation prediction into one linear regression formula, but that still does not beat our predictors.

## 4.4 ROBUSTNESS ANALYSIS

In this experiment, we want check whether our results in the previous section is robust (**Q2**). In our framework, ID does not depend on any hyper-parameters. Therefore, we mainly examine the following hyper-parameters for CL: the number of clusters and the number of neighbors used in the KNN Learner. The results can be found in Figure 3. The resulting linear predictor can still produce a higher correlation than the rotation prediction for a wide range of the configurations we have tried, for both ImageNet and STL-10.

On both datasets, we observe that in general decreasing neighbor number leading to a better predictor. We hypothesize that lowering the neighbor number would results in a stronger KNN classifier that over-fits the structure-less cluster assignments more easily, resulting in better distinction among different checkpoints. For practitioner, we recommend setting number of neighbors to be 1.

On both datasets, we observe that there is an optimal cluster numbers so that the choice of neighbor numbers can be more flexible. If the cluster number is too low, the results become worse. Recall that for different checkpoints, the clusters are produced with the same $K$-means algorithm, so the separability among clusters might be roughly controlled. As a result, we hypothesize that here the cluster numbers here might control the underlying difficulty of the classification problem faced by the KNN learner. In the extreme cases we can have either only 1 cluster or as many clusters as data points. In the former, KNN accuracy would always be 100% and in the latter always be close to 0%. As a result, neither of them would be suitable enough to distinguish the collected checkpoints. As a

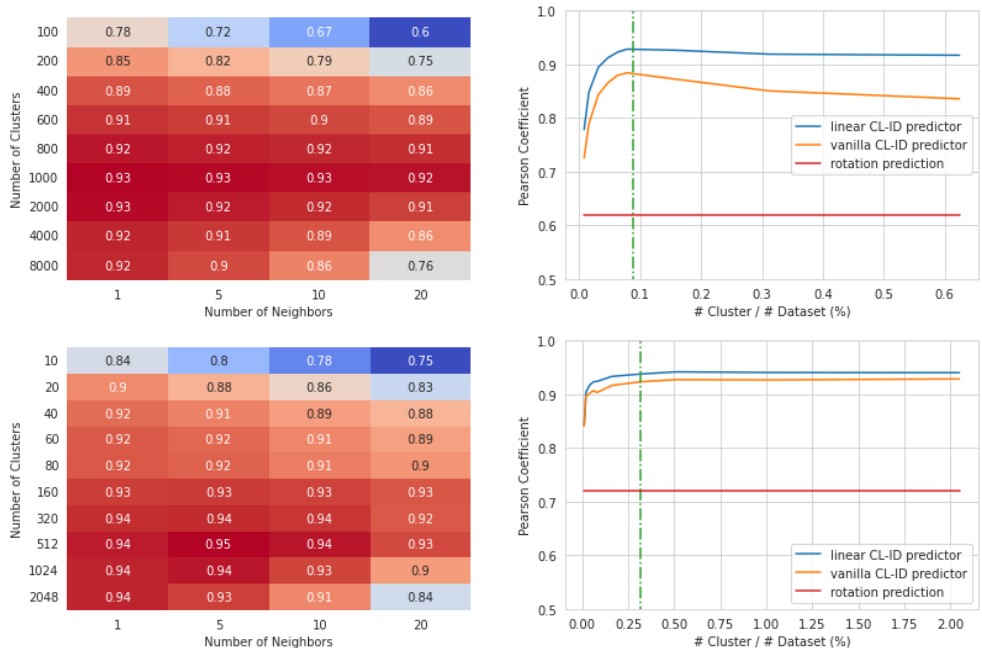

Figure 3: Robustness Analysis for ImageNet (**First Row**) and STL-10 (**Second Row**)). **Left Column**: The heat map of Pearson Coefficient between the Top-1 accuracy and linear CL-ID predictor. **Right Column**: Pearson coefficient vs. the ratio between the number of clusters and the dataset size when using one neighbor. The result is stable with a reasonably large number of clusters, e.g., the square-root of the dataset size (**Green Dashed Line**).

rule of thumb, we recommend using a reasonably large number of clusters, e.g., the square root of the dataset size.

## 5 EXPERIMENT: DEEPCLUSTER WITH IMPROVED CLUSTER LEARNABILITY

In this section, we propose a simple modification for DeepCluster (Caron et al., 2018), which increase the learnability of the representations. Our method is called KNNDeepcluster. The purpose of this experiment is not to focus on a new state-of-the-art SSL method, but rather to understand whether our framework can improve an existing SSL method. The experiments would address the following questions:

**Q3:** Does KNN-DeepCluster out-perform DeepCluster consistently? Does it end up having a higher CL?

**Q4:** Can we predict the performances of the new checkpoints with the CL-ID predictors from section 4.3?

DeepCluster alternates between running $K$-means on the representation space and learning a representation such that it is highly predictive of the target cluster label associated with it. As a result, if we can make target cluster label to be more learnable, we can also make the representation more learnable. Following our approach of CL described above , we will enforce the $K$-means labels to be consistent with the KNN prediction, We proceed in the following way: at the beginning of the epoch, after each $K$-means clustering round, we change the $K$-means labels to the leave-one-out nearest neighbor predictions[2]. The pseudocode of the approach can be found in Algo. 1.

---

[2]In practice, instead of performing a KNN prediction by querying the whole dataset, we use a queue of representations stored in each GPU worker. While this does not result in the exact neighbors, we find that it's more scalable to large datasets.

---

**Algorithm 1:** KNNDeepCluster. $X$ is input images and $\mathcal{F}$ is the model

---

**while** *Not converged* **do**
  Compute $z_i = \mathcal{F}(x_i; \theta)$ for $i = 1, ..., N$
  Kmeans clustering on $\{z_i\}_{i=1}^N$ and get cluster label $\{\tilde{y}_i\}_{i=1}^N$.
  Get $\hat{y}_i = KNNClassifier(z_i|\{z_k, \tilde{y}_k\}_{k \neq i})$ for $i = 1, ..., N$      ▷ Leave one out
  Update $\theta$ on the dataset $\{x_i, \hat{y}_i\}$ for one epoch.
**end**

---

## 5.1 RESULTS ON IMAGENET AND STL-10

| Dataset | STL10 | | | ImageNet | | |
| Eval. Method | Linear | 1NN | 5NN | Linear | Linear 10% | 20NN |
|---|---|---|---|---|---|---|
| SeLa-v2 (Caron et al., 2020) | - | - | - | 71.8 | - | 61.6 |
| DeepCluster (Caron et al., 2020) | - | - | - | **74.3** | - | 65.8 |
| DeepCluster (ours) | 82.9 | 74.3 | 76.6 | 74 | 66.7 | 65.2 |
| KNNDeepCluster (ours) | **83.1** | **76.0** | **79.1** | 74 | **66.9** | **66.6** |

Table 1: Comparison between-KNN0DeepCluster and other cluster-based SSL. The official checkpoints are taken with multicrop of 2x160 + 4x96 and 400 epochs.

In our experiments, we set the number of neighbors to be 3 in our KNNDeepCluster for both datasets, and we find that this does not affect our results much. We perform experiments on both STL-10 and ImageNet. Our results are shown in Table 1. There are consistent gains on both STL10 and ImageNet, but the gain on ImageNet is smaller, and only appears with a weaker downstream classifier such as a KNN or with less training data.

## 5.2 ANALYSIS WITH THE CL-ID FRAMEWORK

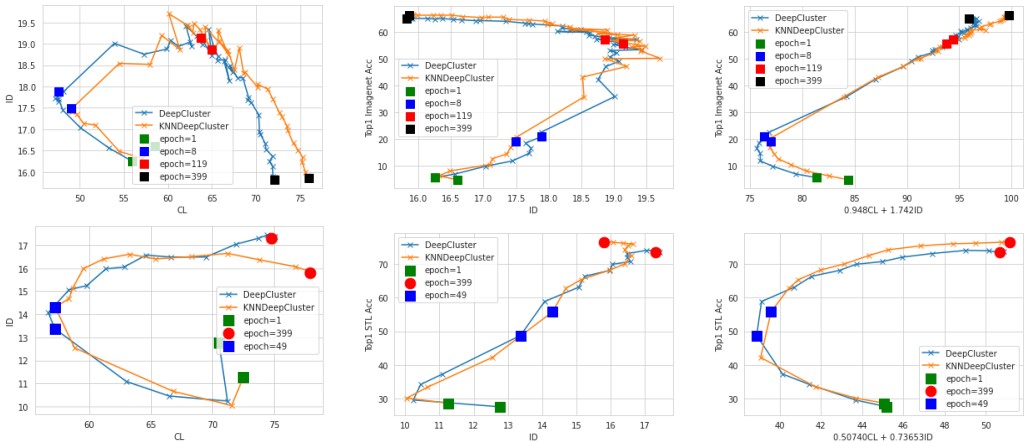

Figure 4: Trajectory of DeepCluster and KNN-DeepCluster for ImageNet (**first row**) and STL-10 (**second row**) for 400 epochs. **Left:** ID vs CL. **Middle:** Top-1 accuracy vs. ID. **Right:** Top-1 accuracy vs. Linear CL-ID predictor (from Section 4.3). Our proposed modification makes the representation more learnable at convergence without a significant compromise of ID, leading to a better solution. There is still a positive correlation between predictors obtained in Section 4.2 and the performance, especially after models are warmed up (after epoch 8 in ImageNet and epoch 49 in STL-10).

We visualize the training trajectories in Figure 4. We observe that the training trajectories for both methods follow similar general patterns across datasets: in the initial stage, the models tend to decrease CL while increasing ID. After the model is properly warmed up, there is an increase of ID

along with CL. Finally, especially for the ImageNet models as well as KNN-DeepCluster in STL-10, ID hits the plateau and starts decreasing while CL keeps increasing. Despite the slight difference in the final stage between two datasets, we find that KNN-DeepCluster converges to a solution that has a higher CL in comparison to DeepCluster, without a significant drop in ID.

We also find that the relative Top-1 classification performance of the intermediate checkpoints can still be well predicted by our linear CL-ID predictor from Section 4.3, especially after the models are property warmed up. The lack of correlation in the early stage checkpoints could be caused by the fact that our linear CL-ID predictor is learnt from the collected 30 checkpoints, which are all about late stage (with Top1 Acc greater than 50%).

Previous work (e.g., Ansuini et al., 2019) claims that ID is inversely correlated to the Top-1 accuracy, seemingly countering our expressiveness argument. Our findings are more nuanced. For ImageNet, at least for the early and middle stages, the ID is increased (before epoch 119) so that, intuitively, the model could explore as much as possible to encode the information from dataset. Then in the later stage, a form of compression occurs while CL increases for better generalization – presumably this stage is the regime of analysis done in Ansuini et al. (2019).

## 6 CONCLUSION & DISCUSSION

Inspired by an analogy with language emergencewe propose to analyze self-supervised representation learning through the lens of expressiveness and learnability. We propose to estimate expressiveness with TwoNN intrinsic dimension (ID), and learnability with the acquisition speed of a KNN learner on the K-means clustering of the representation. Our large-scale evaluation with 30 checkpoints supports our claim. Our method better predicts the top-1 accuracy than other evaluation frameworks, while making less assumptions on data distributions, training algorithms or underlying architectures. According to our method, we modify DeepCluster and our version outperforms it by achieving better cluster learnability on both ImageNet and STL-10. Visualizations show that our framework could predict the performance of the unseen intermediate checkpoints.

We believe our work is a solid step towards understanding the current SSL algorithms and opens up interesting research directions. Future works could further explore the expressiveness-learnability in a more theoretical context, extend this framework to other field like pretrained language models, as well as devise new SSL algorithms that directly maximize intrinsic dimension or cluster learnability.

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

# Appendices

## A   FUTHER DISCUSSION ON CURRENT SSL PRACTICE

We further discuss the motivation behind our framework, and how can approach some of the SSL practice throught the lense of expressiveness and learnability.

**Contrastive Learning**   With our new view on representation quality by expressiveness-learnability, we can try to analyze some of the practice in the contrastive learning and see why it works and fails. Contrastive learning is a typical example of such a trade-off, and it has the following objective:

$$\mathbb{E}_{x,x^+,x^-} \log \Big( \frac{e^{\mathcal{F}(x)^T \mathcal{F}(x^+)}}{e^{\mathcal{F}(x)^T \mathcal{F}(x^+)} + e^{\mathcal{F}(x)^T \mathcal{F}(x^-)}} \Big)$$

It aims to output distinguishable feature vectors between positive and negative examples which turns out to increase the expressiveness. Meanwhile, it aims to make the output vectors among positive examples to be similar by which a higher value of learnability is ensured, since ideally the another learner only need to see one example to generalize to other positive examples. As a result, the degree of learnability here can be controlled by the definition of positive examples. A more diverse selection of the positive examples should make the $\mathcal{F}$ leaning toward the learnability side, since each example could represent a larger fraction of the inputs. It is well acknowledged in the self-supervised community that a diverse data augmentation is an essential part of the contrastive learning, and under this framework, it is a source of the learnability pressure.

**Clustering-based Learning**   Perhaps our expressiveness-learnability framework is more obvious in the clustering-based algorithm like DeepCluster (Caron et al., 2018) and SwAV (Caron et al., 2020). In these algorithms, you learn not only $\mathcal{F}$ but also a bank of cluster centroids $\mathcal{C}$ as well as assignments $\tilde{y} = \mathcal{C}^T \mathcal{F}(x)$. As a result, we can also view these algorithms as finding $\tilde{y} = \mathcal{G}(x)$, where $\mathcal{G}$ is composed of $\mathcal{F}$ and $\mathcal{C}$.

In these methods, the expressiveness is achieved by using a large number of clusters as well as some uniformity constraint, and this can ensure each data has a distinctive cluster assignment. SwAV achieves it with the SinkHorn iterations, while the DeepCluster, at least in the early version, would balance the sampling ratio of images to make the cluster uniform as well as resolving the empty cluster issues.

The learnability requirements are implicitly enforced as well. Since the space of $\tilde{y}$ is usually less than $x$, there are usually multiple data being assigned to the same cluster. This already ensure that a new learner should able to see a few examples to generalize within the cluster. In addition, these algorithms also leverage the data augmentation, so that $\mathcal{G}(x)$ and $\mathcal{G}(x^+)$ should have the same $\tilde{y}$, which further improves the learnability.

## B   CLUSTER LEARNABILITY AND PREQUENTIAL CODELENGTH

Note that Epn. 3 can be connected to the compression lower bound of the emerged cluster assignments datasets $\{z_i, \tilde{y}_i\}_{i=1}^N$ via prequential coding (Dawid, 1984). The prequential codelength is defined as

$$L_{preq} = - \sum_i \log p(\tilde{y}_i | (z_{k \leq i}, \tilde{y}_{k < i})) \tag{4}$$

where $p(\tilde{y}_i | (z_{k \leq i}, \tilde{y}_{k < i}))$ is a conditional model, which can be computed from our KNN classifier probabilities. If Eqn. 3 is the online learning accuracy, then Eqn. 4 is online learning cross-entropy loss. Higher CL lead to lower prequential compression bounds on the emerged clusters.

# C    OTHER ATTEMPTS ON CHARACTERIZING EXPRESSIVENESS AND LEARNABILITY

In this section, we discussed our other unsuccessful attempts on characterizing expressiveness and learnability.

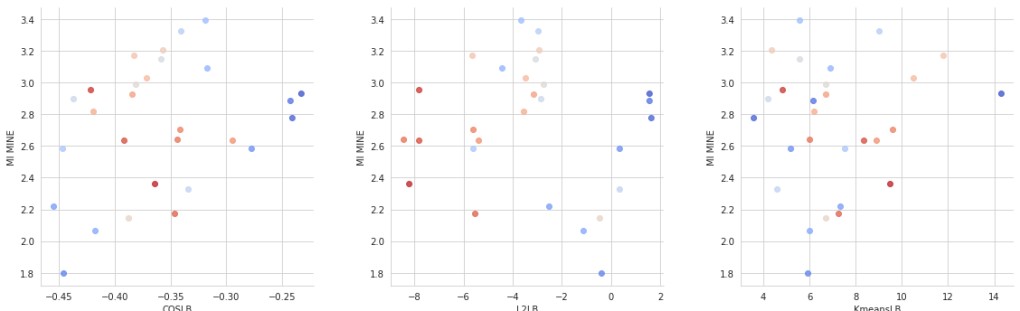

Figure 5: Other attempts on measuring expressiveness and learnability. In all cases, MI is estimated by training a MINE network, where learnability is estimated by training student network that takes the images and predicts the representations with cosine loss (**Left**) or L2 loss (**Middle**) or the $K$-means labels of the representations (**Right**).

Previous works propose maximizing mutual information between inputs and the representations as a way of self-supervised learning (Bachman et al., 2019a; Xue et al., 2020; Hjelm et al., 2018; Linsker, 1988). However, Tschannen et al. (2020) argue that there is a more holistic picture behind their success story, beyond mutual information alone, e.g., the trade-off between the amount of information that can be stored against how hard it is to extract it in the downstream tasks.

We hypothesize that mutual information is indeed part of the solution and it corresponds to the expressiveness of representation. E.g., a one-to-one map between images and representations would have a high MI, while a degenerate solution would not. As a result, we tried using MINE (Belghazi et al., 2018) to compute a lower bound on the mutual information. We use ResNet18 as our MINE network and train for 5 epochs and report the estimate for validation set.

For learnability, we also try fitting a student ResNet18 from scratch to learn the map from images $x_i$ to representations $z_i$ produced by the pretrained checkpoints. Since the representation space is continuous, we experiment with both L2 loss and cosine similarity. Furthermore, inspired by the cluster learnability, we propose to also propose to let the student network to predict emerged cluster labels $\tilde{y}_i$ from images $x_i$. We also use ResNet18 as our student, train for 5 epochs, and report the validation metrics as the learnability.

The qualitative results can be found in Figure 5. We find all these attempts does not produce any meaningful signal. There many problems. Firstly estimating MI usng MINE is known to be sensitive to many things like learning rate and architecture of MINE network (Tschannen et al., 2020). Secondly the loss of directly mapping to the continuous vectors $z_i$ is heavily effected by dimensionality of the space. Thirdly, there is a architecture bias when we use ResNet18 as the students. ViT checkpoints in general performs poorer since it is too different from the ResNets. As a result, our proposed method CL and ID do not have these problem, mainly because we avoid training another neural network from scratch.

# D    MORE RESULTS ON KNNDEEPCLUSTER

We performed more experiments on STL-10 between KNNDeepCluster and DeepCluster under different number of prototypes. The results are in Table 2. We find that our method is able to consistently out-perform the baseline under different configurations.

To further understand the effect of replacing the K-means labels with the KNN predictions, we perform some clustering analysis on a pretrained DeepCluster checkpoint in Figure 6 on STL-10.

| # Prototypes / Eval. Method | | STL10 | |
|---|---|---|---|
| | | 1NN | 5NN |
| 2048 | DeepCluster | 74.96 | 77.78 |
| | KNNDeepcluster | 76.79 | 78.96 |
| 512 | DeepCluster | 74.31 | 76.63 |
| | KNNDeepcluster | 76.03 | 79.11 |
| 128 | DeepCluster | 75.05 | 77.71 |
| | KNNDeepcluster | 75.59 | 78.86 |

Table 2: Performance at different number of clusters.

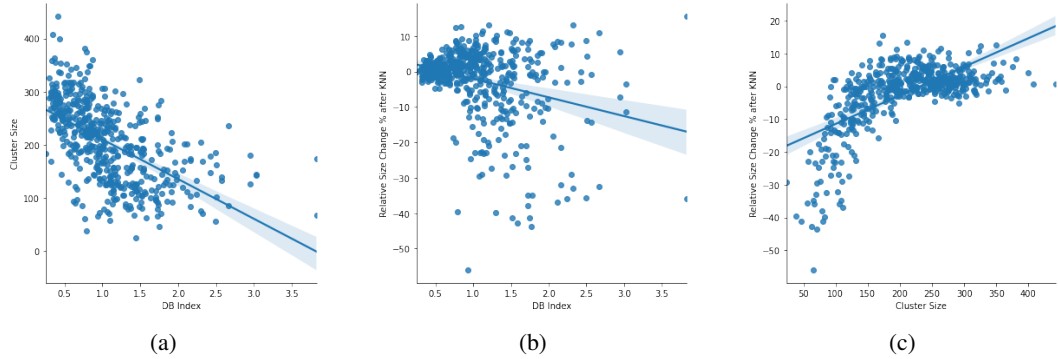

(a)                               (b)                               (c)

Figure 6: Cluster Analysis with Davies–Bouldin (DB) index and the effect of replacing Kmeans labels with KNN predictions on STL10 with 512 clusters. *Left:* After Kmeans, each cluster has unequal size, and the larger cluster are also well separated (low DB index). *Middle and Right:* The originally well separated clusters gain size while the opposite ones lose size after KNN re-assignment. The smallest DB index clusters are least effected by the re-assignment.

We find that after the initial K-means clustering there is an uneven distribution of cluster size, and larger clusters have lower Davies–Bouldin index (DB Index)are more separable from the smaller cluster. We find the reassignment with KNN prediction would punish the smaller clusters with larger DB index, and their cluster members are absorbed into large clusters. As a result, in the next epoch, the network pays more attention on fitting labels coming from large and well-separable clusters

| Name | Method | Architecture |
|---|---|---|
| resnet18 | Supervised | ResNet |
| resnet34 | Supervised | ResNet |
| resnet50 | Supervised | ResNet |
| moco_v1_200ep | Unsupervised | ResNet |
| noco_v2_200ep | Unsupervised | ResNet |
| moco_v2_800ep | Unsupervised | ResNet |
| selav2_400ep_2x224 | Unsupervised | ResNet |
| selav2_400ep | Unsupervised | ResNet |
| deepclusterv2_400ep_2x224 | Unsupervised | ResNet |
| deepclusterv2_400ep | Unsupervised | ResNet |
| deepclusterv2_800ep | Unsupervised | ResNet |
| swav_100ep | Unsupervised | ResNet |
| swav_200ep | Unsupervised | ResNet |
| swav_800ep | Unsupervised | ResNet |
| dino_resnet50 | Unsupervised | ResNet |
| dino_deitsmall16 | Unsupervised | ViT |
| dino_deitsmall8 | Unsupervised | ViT |
| dino_resnet50 | Unsupervised | ViT |
| dino_vitbase16 | Unsupervised | ViT |
| dino_xcit_medium_24_p16 | Unsupervised | XCiT |
| dino_xcit_small_12_p16 | Unsupervised | XCiT |
| simclrv2_r501xsk0 | Unsupervised | ResNet |
| simclrv2_r501xsk1 | Unsupervised | ResNet |
| simclrv1_resnet50_1x | Unsupervised | ResNet |
| insdis | Unsupervised | ResNet |
| pirl | Unsupervised | ResNet |
| infomin | Unsupervised | ResNet |
| pcl_v1 | Unsupervised | ResNet |
| pcl_v2 | Unsupervised | ResNet |
| byol | Unsupervised | ResNet |

Table 3: Full list of the pretrained checkpoints collected. The list of SSL methods are: MoCo-v1 (He et al., 2020), MoCo-v2 (Chen et al., 2020c), SeLA-v2 (Caron et al., 2020), DeepCluster-v2 (Caron et al., 2020), SwAV (Caron et al., 2020), DINO (Caron et al., 2021), SimCLR v2 (Chen et al., 2020b), SimCLR-v1 (Chen et al., 2020a), PCL-v1 (Li et al., 2021), PCL-v2 (Li et al., 2021), PIRL (Misra and Maaten, 2020), BYOL (Grill et al., 2020), InfoMin (Tian et al., 2020), InsDis (Wu et al., 2018).

