# OpenReview forum: "Learnability and Expressiveness in Self-Supervised Learning"
_ICLR.cc/2022/Conference — ICLR 2022 Submitted_

### Official Review · Reviewer_Y1ph · 2021-10-18

**Correctness:** 3
**Technical Novelty And Significance:** 3
**Empirical Novelty And Significance:** 3
**Recommendation:** 5
**Confidence:** 3

**Main Review:**

strengths:

(1) This paper is generally well-written and the motivation is clear. How to better evaluate the representations learned by various self-supervised learning methods is an important question and it will give further instructions to develop new SSL algorithms.

(2) The experimental results demonstrate the effectiveness of the proposed metric CL and ID. I appreciate the authors for their hard but valuable work to collect and evaluate 30 existing SSL checkpoints.

----------------------------------------------------------------------------------------------------------------------------------------------------------

weakness:

(1) It is not clear why we should use CL and ID for evaluation. In other words, is it sufficient (enough) to use these two metrics? I think this work could be better to give some theoretical analysis (on the decomposition of CL and ID) instead of heuristic design. Although the experimental results show some advantages, it is still hard to understand why should we use these two metrics.

(2) The alignment and uniformity framework is more suitable for contrastive methods (by decomposing the InfoNCE loss into two terms) and it may be not an appropriate indicator for other methods, e.g., DINO. Hence, I think the comparison method is not a strong baseline.

(3) I think it is better to design an SSL algorithm by directly optimizing CL and ID. The modifications made on DeepCluster seems a little far-fetched with proposed objective and the results are not convincing enough, especially on ImageNet. It feels to me that it is hard to formalize CL and ID as our loss objectives to optimize. Although mentioned in the conclusion, I think the authors should make more efforts along this direction to make this work more complete.

(4) As the main result of this paper, Figure 1 is blurry. It is not clear what is the relationship between accuracy and colors of different brightness. Also, there exist some typos (e.g., 'deepcluste' should be 'deepcluster' in Figure 1). Hence, I suggest the authors to improve the quality of Figure 1.

**Summary Of The Paper:**

This paper proposes to use Cluster Learnability (CL) and Intrinsic Dimension (ID) to evaluate the representations learned by self-supervised learning methods. The authors collected 30 checkpoints and show that their method is more predictable when compared to other methods, e.g., alignment and uniformity. Moreover, the authors modified the labels generated by K-means in DeepCluster to improve the learnability of the representations and the results demonstrate slight improvements.


**Summary Of The Review:**

This paper is generally well-written and the motivation is clear. However, I still have some concerns (see weakness) and I think this paper could be more complete by providing theoretical analysis or developing an SSL algorithm which directly optimizes CL and ID.

---

> ### Author Response · Authors · 2021-11-12
> **Author Response: Pre-revision**
>
> `As the main result of this paper, Figure 1 is blurry. It is not clear what is the relationship between accuracy and colors of different brightness.`
>
> We are sorry for the confusion. We provide more clarification about Fig 1 in the common response. Red points are checkpoints with high KNN acc while blue points are lower KNN acc. We are striving to improve Fig 1 in our next revision.
>
> `The alignment and uniformity framework is more suitable for contrastive methods (by decomposing the InfoNCE loss into two terms) and it may be not an appropriate indicator for other methods, e.g., DINO. Hence, I think the comparison method is not a strong baseline.`
>
> Yes, we definitely acknowledge the limitation of the Align-Unif framework, and we provide additional insights on why it fails in the common response. Nevertheless, this is to our best knowledge one of the most general evaluation frameworks we could find. If you know some other baselines, please don’t hesitate in letting us know and we are more than happy to add new experiments.
>
> `Although the experimental results show some advantages, it is still hard to understand why we should use these two metrics.  `
>
> The main goal of our paper is not necessarily in convincing people to use CL-ID for actual algorithm development, but we would be happy to know if practitioners find it useful for their SSL algorithm development when there are no labels around.
>
> The main focus of our work here is to present an empirical study, and we believe our work advances understanding representation quality: representations should be expressive (high ID) but compressible/learnable (high CL). To the best of our knowledge, our work is the first to predict downstream performance **without relying on label, architecture, or augmentation.**
>
> `I think this work could be better to give some theoretical analysis and to design an SSL algorithm by directly optimizing CL and ID.`
>
> Yes, we definitely agree! Having theoretical analysis and proposing algorithm that leverages CL-ID in better way than KNNDeepCluster are interesting directions, especially after the intriguingly high correlations we are showing here. However, we believe that these directions are more suitable for future work.

---

### Official Review · Reviewer_xT8d · 2021-10-31

**Correctness:** 3
**Technical Novelty And Significance:** 2
**Empirical Novelty And Significance:** 2
**Recommendation:** 5
**Confidence:** 3

**Main Review:**

General comments:
The general direction and the problem setting which this paper aims to investigate are interesting, but I found that both ID and CL frameworks are heavily identical to the previous frameworks and offer no additional insights on understanding representation quality or other aspects of network learning.

Detailed comments:
-- I found that the writing and notations of this paper are quite hard to follow. And Some claims and analygies are over-exaggerated. For example, i) I don’t understand how language compositionality has anything related to the motivation of this work. ii) In contribution (section 1), the authors said that they analyze self-supervised learning using the proposed methods and then in the second paragraph, they said they evaluates on both self-supervised and supervised models. iii) In Section 3.3, a lot of terms are not properly explained: I don’t understand what $KNN(z_i|z_{k<i},\tilde{y}_{k<i})$ , and “seeing the pairs”, “online learning accuracy” really mean.  iv) I don’t think Q1 and Q2 in Section 4 are different. If high-performing models are having high CL and ID scores, and automatically it indicates that they have a high correlation?

-- Low novelty. The formulation of ID is nearly identical to the original publication from Ansuini, et al, including the explanations and notations used in Section 3.2. The formulation of KNN seems to be a straightforward extension.

-- Comparing to Align-Unif. The results in the original paper indicate that lower alignment and lower uniform would have a better prediction performance. But in the Fig 1., it seems to indicate otherwise, which is not further justified.

-- Using CL framework to improve self-supervised methods leads to very marginal improvement on ImageNet. Without additional information such as variance, it’s hard to confirm that CL systematically improves representation learning. Further, I found this section is a bit deviate from the main focus of this work. Additional experiments, insights or visualisation on how different ID and CL could lead to different representation learning quality and limitation should be more helpful on justifying/understanding “expressiveness” and “learnability” of representation learning.


**Summary Of The Paper:**

This paper proposes Intrinsic Dimension (ID) and Cluster Learnability (CL) as an alternative solution to efficiently evaluate learned representation quality of pre-trained networks, without using down-stream tasks/labels. These two frameworks are inspired by finding “expressiveness” – finding the smallest number of variables to approximate the representation; and “learnability” – number of data samples needed for KNN clustering.

Extensive experiments have been evaluated on models trained with both supervised and unsupervised methods, and results showed that ID + CL would produce a high correlation with prediction accuracy from these methods, comparing to baselines. Finally, the authors argue that CL can also be incorporated into DeepCluster (an unsupervised method), as an auxiliary loss to further improve the prediction performance.


**Summary Of The Review:**

This paper proposed Intrinsic Dimension (ID) and Cluster Learnability (CL) to estimate supervised and self-supervised model's prediction performance. However, the proposed frameworks are heavily identical to the existing works. Considering the less-structured writing and confusing notations, I believe this work has not met the publication quality.

---

> ### Author Response · Authors · 2021-11-12
> **Author Response: Pre-revision**
>
> `Both ID and CL frameworks are heavily identical to the previous frameworks and offer no additional insights on understanding representation quality or other aspects of network learning.`
>
> We respectfully disagree. We understand your concern that the proposed metric is similar to previous works. However, even if we borrow from ID literature, the CL formulation is novel, and the idea of characterizing representational quality in the CL-ID spectrum is also novel. We believe our work advances understanding representation quality: representations should be expressive (high ID) but compressible/learnable (high CL). **To the best of our knowledge, our work is the first to predict downstream performance using label/architecture/augmentation-agnostic metrics.**
>
> `Compared to Align-Unif. The results in the original paper indicate that lower alignment and lower uniform would have a better prediction performance. But in the Fig 1., it seems to indicate otherwise, which is not further justified.`
>
> We are sorry for the confusion. The terms “align” and “unif” in the original paper refers to alignment/uniformity loss. However, we refer to “align” and “unif” as the negated alignment/uniformity loss, which is how we produce Fig.1. With this in mind, we find that Align-Unif does not produce correct results in our context. As is suggested in sec 4.3, this is due to the fact that the architecture constraint of Align-Unif is not satisfied in our setting. More details can be found in the common response.
>
> `Using CL framework to improve self-supervised methods leads to very marginal improvement on ImageNet. Without additional information such as variance, it’s hard to confirm that CL systematically improves representation learning. Further, I found this section is a bit deviate from the main focus of this work.`
>
> Our main goal with this section is to prove that CL-ID framework’s predictive power can be practically extended to new SSL checkpoints, and providing a new SOTA algorithm is only a minor goal. We show that the predictors obtained in section 4.3 could be used to predict the performance of new SSL checkpoints, especially after models are warmed up. This demonstrates the potential of using the CL-ID beyond existing SSL checkpoints. Please see the common response for more details.
>
> `Additional experiments, insights or visualisation on how different ID and CL could lead to different representation learning quality and limitation should be more helpful in justifying/understanding “expressiveness” and “learnability” of representation learning.`
>
> Yes, we definitely agree! Especially since we empirically show such intriguingly high correlation with downstream accuracies. Nevertheless, our main focus for this paper is to establish our findings with large-scale experiments. Therefore, we think the theoretical insights and understanding are more suitable for future work.
>
> `I don’t understand how language compositionality has anything related to the motivation of this work.`
>
> We are sorry for the confusion. We plan to refactor our introduction section focusing on representation evaluation, and we will move the link to language emergence into appendix for interested readers.
>
> `In contribution (section 1), the authors said that they analyze self-supervised learning using the proposed methods and then in the second paragraph, they said they evaluates on both self-supervised and supervised models`
>
> To illustrate the generality of the approach, we do not limit ourselves to self-supervised and aim to find a unifying evaluation framework that can also incorporate supervised learning. We believe the fact that our method works for both SSL and supervised models is a merit. We can also provide the results excluding the supervised checkpoints in the revision if you think this is confusing.
>
> `In Section 3.3, a lot of terms are not properly explained`
>
> We are sorry for the confusion. We now provide more details and intuition on the computation of CL. After clustering, we have a dataset labelled with cluster assignments $\\{z_i, \tilde y_i \\}$. We adopt an online learning scenario here. That is, our KNN classifier only goes through the dataset once. As a non-parametric classifier, KNN maintains a memory of previously seen data points, and for the new data, KNN finds the neighbors in the memory to predict cluster labels. We use $\hat y_i = KNN(z_i | z_{<i}, \tilde y_{<i})$ to denote the KNN prediction on data $z_i$, where the neighbors are from the previous $i-1$ data points. At the beginning $i=1$, the KNN is outputting random predictions. With $i$ increasing, the prediction becomes more accurate since the KNN has seen more and more examples. Therefore, we propose to measure the learning speed with the online learning accuracy, which is the average prediction accuracy for $i$ from $1$ to $N$.

---

> > ### Comment · Reviewer_xT8d · 2021-11-29
> > **Final response**
> >
> > I agree with the motivation of this work, developing architecture-agnostic solutions to rank self-supervised methods is a very interesting research direction. However, the proposed solution of ID/CL framework does not properly justify this.
> >
> > Having the similar concerns from Reviewer Brvt, it's not clear for me the correlation between ID/CL and prediction accuracy. And it's not clear what's the underlying meaning of "expressiveness" and "learnability", and how we can use the ID/CL to understand the quality of learned representations.
> >
> > From the rebuttal, some of my confusion on the notations and baselines have been clarified. However, considering the fact that there are some major issues regarding the method design are unclear, I would maintain my original rating.

---

### Official Review · Reviewer_Fbk6 · 2021-11-02

**Correctness:** 3
**Technical Novelty And Significance:** 2
**Empirical Novelty And Significance:** Not applicable
**Recommendation:** 5
**Confidence:** 4

**Main Review:**

Advantage:
1.	Current assessment methods for self-supervised learning depend on the availability of labels, while the two metrics proposed in this paper escape this dependency. The proposed metrics provide a new evaluation protocol for self-supervised learning in the case where annotation is absent.
2.	The proposed two metrics are shown to be highly correlated with the performance of the self-supervised methods on the KNN classification task, which inspires further exploration of the relationship between expressiveness-learnability and representation quality.

Disadvantage:
1. The performance of linear evaluation and KNN classification tasks on the pre-trained dataset is approaching a bottleneck, and more attention has been shifted to the transferred performance on other datasets. However, this paper only shows the correlation between classification performance on the pre-trained dataset and the proposed metrics, while ignoring the transfer performance on other datasets, which limits the impact and significance of the paper. The current application scenario is not attractive for unsupervised learning.
2.  Complexity is important for the scope of application of the method. However, the complexity of the proposed metrics is not discussed in this paper.


**Summary Of The Paper:**

This paper proposes two metrics for assessing the quality of self-supervised learning representations in terms of expressiveness and learnability. Expressiveness expects to maximize the mutual information between the representation and the original data, while learnability emphasizes the representation suppose to be simpler and more learnable. Based on these two intuitions, the author applies Intrinsic Dimension (ID) and Cluster Learnability (CL) as the measurements of expressiveness and learnability respectively to predict downstream classification performance. The authors collect 30 checkpoints of recent self-supervised methods to validate the proposed metrics on two datasets, ImageNet and STL_10, and the results show a high correlation between the proposed metrics and the downstream classification performance.

**Summary Of The Review:**

1. Section 3.2 indicates that the estimator is exact for uniformly distributed data, and datasets applied in experiments are balanced. Is the proposed metrics only work for the balanced dataset?
2. Figure1 is fuzzy and the name of methods overlap badly, which affects the reading. The description to the calculation of CL is unclear and confusing. If I understand correctly, is each sample is classified according to all the samples have seen before?
3. The improvement brought by KNNDeepCluster is tiny compared to the base method, which is hard to be convinced of the effectiveness of the proposed method.

---

> ### Author Response · Authors · 2021-11-12
> **Author Response: Pre-revision**
>
> `The performance of linear evaluation and KNN classification tasks on the pre-trained dataset is approaching a bottleneck, and more attention has been shifted to the transferred performance on other datasets.`
>
> We agree! Our experiment on STL suggests the transferability of our CL-ID framework. In our experiment, we reuse the checkpoints pretrained on ImageNet, and compute ID and CL with STL-10 images. Figure 1 suggests you can use CL-ID to predict performance on new datasets, as long as you can obtain images from new datasets. Due to the time limit of the rebuttal, we might not be able to add other tasks like object detection, but we are striving to add some other image classification tasks.
>
> `Complexity is important for the scope of application of the method. However, the complexity of the proposed metrics is not discussed in this paper.`
>
> The computation complexity of ID and CL is similar to that of the KNN evaluation, where the main bottleneck is computing the representations of each image and the distance matrix. In our work, we pre-computed these representations. We will add this consideration to the paper.
>
> `Section 3.2 indicates that the estimator is exact for uniformly distributed data, and datasets applied in experiments are balanced. Does the proposed metrics only work for the balanced dataset?`
>
> We share the reviewer's concern for the dataset imbalance, but as is pointed out by Ansuini et al. the TwoNN estimator remains asymptotically correct for non-uniform sampling for large enough datasets.
>
> `Figure1 is fuzzy and the name of methods overlap badly, which affects the reading. `
>
> We provide details about Figure 1 in the common response, and we are sorry for the confusion.
>
> `The description of the calculation of CL is unclear and confusing. If I understand correctly, is each sample classified according to all the samples I have seen before?`
>
> Yes, your understanding of CL is correct. We provide more details and intuition on the computation of CL. After clustering, we have a dataset labelled with cluster assignments $\\{z_i, \tilde y_i \\}$. We adopt an online learning scenario here. That is, our KNN classifier only goes through the dataset once. As a non-parametric classifier, KNN maintains a memory of previously seen data points, and for the new data, KNN finds the neighbors in the memory to predict cluster labels. We use $\hat y_i = KNN(z_i | z_{<i}, \tilde y_{<i})$ to denote the KNN prediction on data $z_i$, where the neighbors are from the previous $i-1$ data points. At the beginning $i=1$, the KNN is outputting random predictions. With $i$ increasing, the prediction becomes more accurate since the KNN has seen more and more examples. Therefore, we propose to measure the learning speed with the online learning accuracy, which is the average prediction accuracy for $i$ from $1$ to $N$.
>
>
> `The improvement brought by KNNDeepCluster is tiny compared to the base method, which is hard to be convinced of the effectiveness of the proposed method.`
>
> As is mentioned in the common response, the utility of the CL-ID framework is mainly demonstrated through our framework’s predictions on the unseen checkpoints, while proposing a new SSL algo is a minor goal here.

---

### Official Review · Reviewer_Brvt · 2021-11-08

**Correctness:** 2
**Technical Novelty And Significance:** 2
**Empirical Novelty And Significance:** 2
**Recommendation:** 5
**Confidence:** 3

**Main Review:**

Strengths:
-  Pearson correlation coefficient between the proposed factors (ID and CL) and final accuracy is surprisingly high. Thus, they can be a very good unsupervised monitor for self-supervised training.
- The proposed algorithm inspired by CL outperforms DeepCluster.

Weakness:
- The technical part is somewhat simple and lacks insight into the proposed factors. For example, the proposed factors seem to be independent of the learning methods and learning paradigms, since they characterize the expressiveness and learnability of pre-trained representations. However, in the top left of Fig 1, DINO has both better ID and CL than resnet50, while supervised learned resnet50 should outperform DINO. As a comparison, for Align-Unif Predictor (top right of Fig 1), resnet50 has better alignment and uniformity than dino (and all self-supervised methods), which matches the fact.
- According to the paper, higher ID indicates better expressiveness. However, in the experiments (top left and bottom left of Fig 1), resnet50 has less ID than resnet34, and resnet34 has less ID than resnet18, which is an apparent contradiction. Therefore, the motivation for seeking higher ID is not well supported.
- Furthermore, Pearson correlation coefficient as the only metric is not enough to conclude that ID and CL are better than other existing predictors. This paper needs more investigation on their scope of application and limitations.

Minor comments:
- Different models have different normalization of the feature. For example, SimCLR forces $||f||=1$ and BYOL does not. Thus, using cosine distance to infer ID is not correct, since cosine distance is equivalent to Euclidean distance iff $||f||=const$.
- Fig 1 is not a vector diagram (become fuzzy when zooming in) and there are many repetitive labels.


**Summary Of The Paper:**

This paper tries to give a measurement method to evaluate the learned model, which is highly correlated with the final test accuracy.
The measurement method depends on two key factors: Intrinsic Dimension (ID) and CLuster Learnability (CL).
This paper claims that the model with higher ID and CL performs better. Using ID and CL to predict top-1 accuracy can achieve a Pearson correlation coefficient of 0.93, which is better than existing predictors (e.g., alignment and uniformity).
Inspired by the above observation, this paper proposes a modified DeepCluster algorithm to increase the final performance.

**Summary Of The Review:**

This paper is novel and the results are interesting. However, the technical part is somewhat simple, and the insight behind the observations needs further exploration.

---

> ### Author Response · Authors · 2021-11-12
> **Author Response: Pre Revision**
>
> `Comment: DINO has both better ID and CL than resnet50, while supervised learned resnet50 should outperform DINO.`
>
> As is mentioned in the common response, the points with the text “dino” come from different checkpoints (See Table 3). While dino_resnet50 is on par with the supervised resnet50, dino_vitbase16 is better than resnet 50. As a result, some points with text “dino” end up having both higher ID and CL, while others only have higher CL with a slight drop on ID.
>
> `Comment: for Align-Unif Predictor (top right of Fig 1), resnet50 has better alignment and uniformity than dino (and all self-supervised methods), which matches the fact.`
>
> Unlike the original paper, on Align / Unif, the good model is supposed to be on the top right corner (we choose to plot the negated alignment and uniformity loss). Therefore resnet50 actually has lower alignment and uniformity than other SSL checkpoints in our setting. We are sorry for the confusion and will clarify it in the revision.
>
> `According to the paper, higher ID indicates better expressiveness. However, in the experiments (top left and bottom left of Fig 1), resnet50 has less ID than resnet34, and resnet34 has less ID than resnet18, which is an apparent contradiction. `
>
> Our full claim is that a good model needs to have both high ID and high CL. For these three data points (resnet18, resnet34, renet50), even when the ID goes down, the CL goes up, so it could still end up a better model, depending on the exact trade-off. To illustrate the role of ID, if you focus on the vertical line of $CL\approx 85$ for imageNet, you can see higher ID leads to better performance when CL is controlled.
>
> `Pearson correlation coefficient as the only metric is not enough to conclude that ID and CL are better than other existing predictors. This paper needs more investigation on their scope of application and limitations.`
>
> We agree that correlation coefficient is not enough. As is mentioned in the common response, this is exactly why we provide a study case with KNNDeepCluster to investigate whether CL-ID framework can be extended to the scope of new SSL checkpoints. Would you have ideas on how we can strengthen this investigation?
>
> `Different models have different normalization of the feature. For example, SimCLR forces ||f||=1 and BYOL does not. Thus, using cosine distance to infer ID is not correct, since cosine distance is equivalent to Euclidean distance iff ||f||=const.`
>
> We found no result difference between using cosine distance or L2 distance. We end up choosing cosine so that we are consistent with the common practice. We would update the results with L2 distance.
>
> Please feel free to suggest new baselines, visualization, and analysis. We are more than happy to include new experiments for the revision.

---

> > ### Comment · Reviewer_Brvt · 2021-11-29
> > **Further Comments**
> >
> > Thanks for the clarification. Your response to comments 1, 2, 5 addresses my concern.
> >
> > However, I am not convinced by your response to comment  3 (the understanding of ID).
> >
> > The motivation of this work is that a good model should have good expressiveness and learnability and vice versa.
> > Then ID and CL are used to be the surrogate for the expressiveness of the representation and cluster learnability, respectively.
> > After that, the empirical observations show that both high ID and high CL indicated a good model.
> > Furthermore, when fixing CL, models with higher ID have better performance.
> >
> > Since different architectures may be hard to compare their expressiveness, we can choose the same architecture with different layers to consider (i.e., supervised trained ResNet-50, 34, 18).
> > Certainly, ResNet-50 has better expressiveness than ResNet-34, and ResNet-34 has better expressiveness than ResNet-18.
> > If we think a better expressiveness leads to a better model, then
> > ResNet-50 should have higher KNN acc than ResNet-34, and ResNet-34 should have higher KNN acc than ResNet-18,
> > Indeed, it is consistent with the fact (fig1).
> > In contrast, in the above case, ResNet-50 has less ID than ResNet-34, and ResNet-34 has less ID than ResNet-18.
> > Thus, how can we claim that ID is a good surrogate for expressiveness?
> >
> > Based on the experimental observations, maybe you can directly claim that the ID instead of expressiveness is the key factor to evaluate a  model.
> > The difference is that expressiveness only depends on the function space $\mathcal F$, while ID depends on the specific learned function in the function space $f\in \mathcal F$.
> >
> > Overall, I think this paper brings some new insights, and I would like to increase my score to 5. However, it still needs more justifications for ID and CL, both theoretically and empirically, which are somewhat thin in the current version.

---

> > > ### Author Response · Authors · 2021-11-29
> > > **Further Clarification on Expressiveness of ResNet18, ResNet34 and ResNet50.**
> > >
> > > Thanks for the response and we appreciate for your recognition! We hereby address your concerns.
> > >
> > > ```
> > > Comment: Certainly, ResNet-50 has better expressiveness than ResNet-34, and ResNet-34 has better expressiveness than ResNet-18.
> > > ```
> > > We respectfully disagree with it. We want to clarify that our usage of expressiveness **does not refer to the capacity of the model class $\mathcal{F}$, but the degree to which the learned function $f\in\mathcal{F}$ can discriminate concepts**. As a result, while ResNet50 might have the tendency to become higher expressiveness, it is not guarenteed. If the learning is improper, ResNet50 could even become zero expressiveness solution (e.g., a degenerate representation that outputs constant vectors for each image).
> > >
> > > ```
> > > Comment: ResNet50 has higher KNN than ResNet34, but lower ID.
> > > ```
> > > Since our main claim is "a good representation has both high ID and high CL", it is not that surprise that the model could still improve by trading off betwee CL and ID. Under our framework, the fact that ResNet-50 has higher KNN and lower ID would predict ResNet-50 has higher CL than the rest of them, which is exactly consistent with our observation. In fact, our KNNDeepCluster also converge to a overally better solution (higher KNN) by increasing CL without compromising too much ID. However, we agree that future algorithm should strive for improving both at the same time for maximal performance gains.
> > >
> > > ```
> > > Comment: Why ID is a good surrogate for expressiveness?
> > > ```
> > > We agree with the reviewers that ID might not be the best surrogate, but it is a reasonable one that satisfy the high-level intuition. Our intuition is: **As the representation becomes more expressive (definition above), it could represent images into more fine-grained categories, which could have multiple disentangled features such as color, shape, positions in space, etc.** If these disentangled features correspond to different directions in some non-linear manifold, then ID should yield a good surrogate for the expressiveness of the representation. As the first work to establish such a framework, the choice of ID already yields strong results even with only high-level intuition. Future work could better investigate a more theoretical surrogate of expressiveness.
> > >
> > > Finally, we again appreciate you raise the score, but hopefully our response above could further clarify the definition of expressiveness, our choice of ID, and the interpretation of the ResNet 18/34/50 results. We would love to hear more from you.

---

> > > > ### Comment · Reviewer_Brvt · 2021-11-29
> > > > **Further Questions**
> > > >
> > > > Thanks for your further clarification.
> > > >
> > > > I am still wondering 1) the principle behind the effectiveness of ID and CL, and 2) the limitation of the ID-CL predictor.
> > > >
> > > > It seems that the value of ID or CL does not depend on the training methods and only depends on the learned encoder and input data. What about the pretext-based methods? For example, image colorization [1], predicting image rotations [2].
> > > >
> > > > Or ID-CL only works for contrastive-based methods and cluster-based methods? Why?
> > > >
> > > > How will the data augmentation affect ID and CL? Are they still highly correlated to the downstream performance for different kinds of data augmentations? Why?
> > > >
> > > > ---
> > > >
> > > > [1] Richard Zhang et al. “Colorful image colorization”. 2016
> > > >
> > > > [2] Spyros Gidaris et al. “Unsupervised representation learning by predicting image rotations”. 2018

---

> > > > > ### Author Response · Authors · 2021-11-29
> > > > > **Author Response**
> > > > >
> > > > > Thanks for the questions. We definitely agree that further explore the principle behind ID-CL and its limitations are interesting questions, but we feel that asking all of them might be too much for one publication, especially with a hard question like representation evaluation. Our main purpose is to first establish such framework, and make sure it is **robustly effective on the main stream SOTA methods**.
> > > > >
> > > > > We appreciate your suggestion on adding the pretext methods checkpoints, and we don't believe it would change the results much, since our framework does not take any assumption on learning algorithm being contrastive or cluster-based. We are in the process of adding these two checkpoints and strive to reach the deadline as the discussion stage ends.

---

### Author Response · Authors · 2021-11-12
**Common Response to All Reviewers**

We use the following abbreviation: R1 (Brvt), R2 (Fbk6), R3 (xT8d) and R4 (Y1ph)

We thank all reviewers for the constructive insights. We appreciate that the reviewers find the our paper to be well-written (R4), our problem to be well-motivated (R3, R4), the proposed CL-ID framework to be surprisingly effective (R1, R2, R4), and our experimental efforts to be valuable (R4). Here we want to address some common concerns, individual responses can be found under each thread. The paper revision will soon follow.
### R1, R2, R3, R4: What are your novelties / contributions?
We propose CL-ID, a representation evaluation framework for pretrained vision checkpoints and provide a large-scale empirical study. While being conceptually simple, easy to implement and efficient to compute, CL-ID could predict the representation quality **without relying on human label / model architecture / learning algorithms / data augmentation, which all the prior works fall short of.** While we partially borrow from the existing literature of Intrinsic Dimensionality (ID), our formulation of Cluster Learnability (CL) and the characterization of representation quality with the CL-ID framework are novel. We believe our work advances understanding representation quality: representations should be expressive (high ID) but compressible/learnable (high CL).
### R1, R2, R3, R4: What are your theoretical insights? How to develop new SSL algorithms leveraging CL-ID?
We agree that having theoretical insights and proposing new SSL algorithms that optimize CL-ID are interesting research directions, especially after the intriguingly high correlation we show here. However, as the first paper in this line of work, our primary focus is to properly establish the framework as well as to show its promise with the large-scale empirical study.
### R1, R2, R3, R4: Figure 1 is fuzzy and confusing, and it seems contradictory to the main claim. Could you provide more details on Align / Unif? Why does it fail?
We apologize for the confusion caused by Fig 1. In Figure 1, each data point is a different checkpoint, while the red points are models with higher KNN acc and blue points are lower KNN acc. For the text annotation, we denote each point with the prefix of that checkpoint, e.g., “swav_100ep” -> “swav”, “resnet50->resnet50”. This explains why there are repetitive points. They are not inconsistent results, but checkpoints with the same prefix. The full names of each checkpoint can be found in Table 3. We will update Figure 1 with vector format, clearer annotation and color bar.

We further clarify our observations on Align-Unif. In the original work [1] the authors plot the alignment loss vs uniformity loss so the best models are expected to be in the lower left corner. However, in our plot, **we choose to negate both losses in our plot, so the best models should appear in the upper right corner, which is consistent with our CL-ID visualization**. Therefore, in our paper “high align / unif” corresponds to “low align / unif loss” in the original work. Therefore Align / Unif produces qualitatively incorrect results in our settings (e.g. DINO are high unif and low align, while supervised resnets are low in both quantities). We are sorry for the confusion and will make it clearer.

As is illustrated in Sec 4.3, the failure of Align / Unif is because their architecture constraint cannot be satisfied in our large-scale experiment settings. To be specific, let the network be $f$, and the projection head be $g$. Then let $z = f(x), o = g(z)$. The alignment and uniformity are defined with $o$ and $o’$ by decomposing the contrastive learning loss. However, for most of our checkpoints, you only have $f$ and no $g$. Therefore we can only attempt to apply the Align/Unif on $z, z’$. Hence our results are not contradictory to the work of [1], but show their limitation when the architectural constraint is not satisfied.
### R2, R3, R4: KNNDeepCluster only leads to marginal improvement. How does it demonstrate the utilities of the CL-ID framework?
We agree with reviewers that KNNDeepCluster only provides incremental improvements for now, however, providing a new SOTA algorithm is only a minor goal here. The main purpose of KNNDeepCluster is to demonstrate the generalization power of the CL-ID when applying to new SSL checkpoints. As is shown in section 5.2, the proposed framework correctly reflects KNNDeepCluster to have a higher CL. In addition, using the predictors obtained in section 4.2, the performance of new checkpoints are still well predicted. This further shows the potential of the proposed CL-ID framework beyond correlation with existing methods. We will rewrite this section to make this motivation behind clearer.

Reference:
[1] Tongzhou Wang and Phillip Isola. Understanding contrastive representation learning through alignment and uniformity on the hypersphere. In International Conference on Machine Learning, pages 9929–9939. PMLR, 2020.

---

> ### Author Response · Authors · 2021-11-16
> **Reminder for Reviewers on Pre-revision Responses**
>
> Dear Reviewers,
>
> It has been a couple of days since we posted our initial rebuttal, and we would love to hear your feedback.
>
> We would also like to remind all reviewers about our robustness analysis in section 4.4, which seems to be overlooked in the comments so far. This is also one of our main contributions. The robustness suggests the CL-ID framework is not only practically useful, but also likely to open doors for interesting theoretical insights for future work, since it’s not a random success. Please don’t hesitate to reach us if you have any other questions!

---

### Decision · Program_Chairs · 2022-01-20

**Decision:**

Reject

**Comment:**

The problem studied in this paper is interesting and the high-level motivation of the proposed research is reasonable. However, as pointed out by reviewers, it is not convincing that the developed components in the proposed method are able to address the issues mentioned in the high-level motivation. Furthermore, the experimental results are not convincing to verify the motivations either. Though the authors provided some clarifications in the rebuttal, reviewers' major concerns still remain.

The authors are encouraged to take reviewers' concerns into consideration to revise the proposed method to make it a stronger work for future submission. Based on its current form, this work is not ready for publication at ICLR.